# VideoMAETok: Boosting Video Diffusion Models via Masked Autoencoders as Tokenizers

Zhan Tong[1]   Tinne Tuytelaars[1]

## Abstract

Latent diffusion models have become the dominant paradigm for video generation, making the video tokenizer a critical component. While most existing tokenizers are trained primarily for reconstruction, diffusion models are optimized to *denoise* heavily corrupted latents, which creates a mismatch between tokenizer training objectives and downstream generative learning. As a result, reconstruction metrics (e.g., rFVD) can be a poor proxy for generation quality (gFVD), and overly prioritizing reconstruction may even hinder diffusion training. We propose VideoMAETok, a simple family of ViT-based video tokenizers trained explicitly as *corruption-inversion* models for latent video diffusion. VideoMAETok builds on masked autoencoders: we (i) apply high-ratio token masking and encode only visible spatiotemporal tokens for efficiency, and (ii) corrupt latent tokens with interpolative Gaussian noise to better match the denoising nature of diffusion generators. Training under such corruption encourages latents that remain informative and well-conditioned for downstream denoising. Extensive experiments show that VideoMAETok consistently improves generation quality when paired with off-the-shelf diffusion models (SiT and LightningDiT), achieving state-of-the-art gFVD on Kinetics-600 and UCF-101 while remaining compute-efficient. Code is available at https://github.com/yztongzhan/VideoMAETok.

## 1. Introduction

The generation of video content has become a major frontier in computer vision, driven by the growing demand

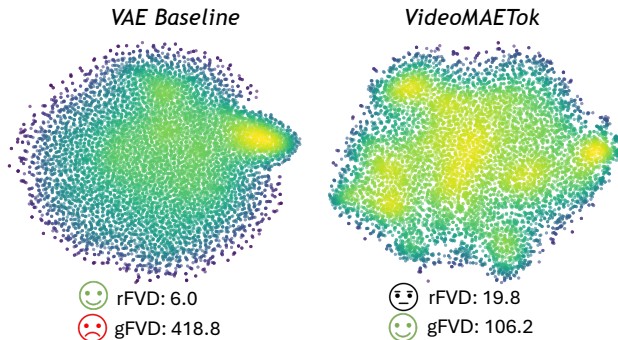

*Figure 1.* **Latent Distribution Visualization of the Tokenizers.** We visualize the latent distributions learned by different tokenizers using t-SNE. Compared to a reconstruction-oriented VAE baseline (the first row in Table 2a) trained on clean inputs (left), VideoMAETok (right) learns a more structured and well-spread latent space by training to reconstruct videos from *corrupted* latents. With the same ViT backbone and the same SiT-B generator, VideoMAETok substantially improves gFVD from 418.8 to 106.2, demonstrating that tokenizer training objectives can directly shape downstream video generation quality.

for realistic, controllable, and high-fidelity video synthesis. Among the many generative paradigms, latent diffusion models (Brooks et al., 2024; Wiedemer et al., 2025; Kong et al., 2024; Wan et al., 2025) have recently emerged as a dominant paradigm for video generation. A key reason is their ability to learn generative dynamics in a compact latent space rather than directly in pixel space (Hoogeboom et al., 2023), which would otherwise be prohibitively expensive. The quality of a latent video diffusion model, however, depends critically on the *video tokenizer* that maps high-dimensional videos into latents. Importantly, a good tokenizer is not only a compressor: it defines the representation space in which the diffusion model must denoise and generate. This creates two desiderata that are often conflated in practice: (i) the tokenizer should preserve visual content to enable faithful decoding, and (ii) its latent space should be *well-conditioned for denoising-based generative learning*. Most existing video tokenizers (Li et al., 2025; Xing et al., 2025; Tang et al., 2024) focus primarily on reconstruction objectives, without explicitly accounting for the fact that diffusion models are trained to invert *corruptions* (noise, partial information, and uncertainty) in latent space.

---

[1]Department of Electrical Engineering (ESAT), KU Leuven, Belgium. Correspondence to: Tinne Tuytelaars <tinne.tuytelaars@esat.kuleuven.be>.

*Proceedings of the 43rd International Conference on Machine Learning*, Seoul, South Korea. PMLR 306, 2026. Copyright 2026 by the author(s).

Recent work in the image domain has highlighted a similar tension: improving reconstruction fidelity does not necessarily improve (and can even harm) downstream generative performance (Yao et al., 2025). In video, this mismatch is amplified by long spatiotemporal sequences and the limited understanding of how Transformer-based tokenizers should be trained for diffusion-based video generation, compared to the well-established convolutional VAE recipe (e.g., SD-VAE (Rombach et al., 2022)). This raises a basic but under-explored question for latent video diffusion: *What makes a video tokenizer good for diffusion-based video generation?*

In this work, we introduce VideoMAETok, a family of ViT-based video tokenizers designed to align tokenizer training with the denoising nature of latent diffusion models (Blattmann et al., 2023; Ma et al., 2024; Yao et al., 2025). Our motivation is simple: since diffusion models learn to reconstruct clean signals from corrupted latents, a tokenizer should be trained under *the same kind of difficulty*—namely, reconstructing clean videos from *strongly corrupted* latent tokens. Concretely, VideoMAETok combines two complementary corruptions: (1) MAE-style high-ratio token masking, which forces the representation to be predictive and robust under missing information, and (2) interpolative latent noise injection, inspired by latent denoising tokenizers (Chen et al., 2024; Yang et al., 2025), which encourages stability under a continuum of noise magnitudes. In addition, unlike VideoMAE (Tong et al., 2022; Wang et al., 2023) that emphasizes a heavy encoder for discriminative tasks (Kay et al., 2017; Goyal et al., 2017; Gu et al., 2018), VideoMAETok adopts a lightweight encoder and a heavier decoder, allocating capacity to faithful decoding and generation-oriented tokenization.

Despite its conceptual simplicity, VideoMAETok yields a strong practical payoff. Across Kinetics (Kay et al., 2017; Carreira et al., 2018) and UCF-101 (Soomro et al., 2012), and with off-the-shelf diffusion generators (e.g., SiT (Ma et al., 2024)), we observe that reconstruction-oriented tokenizers can be misleading, whereas corruption-aware training consistently produces latents that substantially improve gFVD. Our results establish VideoMAETok as a strong, easy-to-adopt tokenizer baseline for latent video diffusion. Our contributions can be summarized as follows:

- We identify and empirically characterize a systematic mismatch between reconstruction-focused tokenizer objectives and downstream video diffusion generation quality, showing that better reconstruction fidelity does not necessarily yield better generation quality.

- We propose VideoMAETok, a simple corruption-aware video tokenizer that combines high-ratio masking and latent noising, tailored for video generative modeling.

- We demonstrate that, when paired with standard diffu-

sion generators such as SiT (Ma et al., 2024) or LightningDiT (Yao et al., 2025), VideoMAETok can achieve state-of-the-art gFVD on UCF-101 and Kinetics-600, establishing a strong, simple baseline and a set of design principles.

## 2. Related Work

### 2.1. Tokenizers for Video Generation

Modern video generation systems typically rely on an auto-encoding tokenizer to map high-dimensional videos into a compact latent space (Brooks et al., 2024; Wiedemer et al., 2025; Kong et al., 2024; Wan et al., 2025). This design is motivated by the sheer scale of video data: operating directly in pixel space is often prohibitively expensive in compute and memory, and makes long-range spatiotemporal modeling difficult. Consequently, most practical pipelines first learn a video-to-latent mapping with an autoencoder, and then train a generative model in the latent space, analogous to the image setting (Rombach et al., 2022; Kingma & Welling, 2013). Existing video tokenizers are largely optimized for reconstruction fidelity and perceptual quality, using either continuous latents (Kong et al., 2024; Wan et al., 2025; Hansen-Estruch et al., 2025) or discrete latents (Yu et al., 2024a; Liu et al., 2025). While strong reconstruction is important for faithful decoding, it does not automatically imply that the latent space is well-suited for downstream generative learning. In fact, prior work has observed that improving reconstruction can be weakly correlated, or even negatively correlated, with diffusion generation quality (Yao et al., 2025). This motivates a central question of our work: *what training objective produces a latent space that is well-conditioned for denoising-based video generation, beyond reconstructing clean inputs?* We address this gap by explicitly training a tokenizer under strong latent corruption, so that its representation remains informative under the same type of difficulty faced by diffusion generators.

### 2.2. Video Generative Models

Visual generative modeling has been studied extensively, with two dominant paradigms: diffusion models (Ho et al., 2020; Rombach et al., 2022; Dhariwal & Nichol, 2021; Peebles & Xie, 2023; Ma et al., 2024) and autoregressive models (Chang et al., 2022; Sun et al., 2024; Yu et al., 2024b; Tian et al., 2024). The same taxonomy extends to video generation, where diffusion-based approaches iteratively denoise all frames to produce high-quality videos (Ho et al., 2022; Blattmann et al., 2023), while autoregressive approaches predict frames or frame chunks sequentially for streaming generation (Yu et al., 2024a; Liu et al., 2025; Teng et al., 2025). Despite differences in the generative paradigm, most modern video generators do not operate directly on raw pixels. Instead, they learn and sample in a

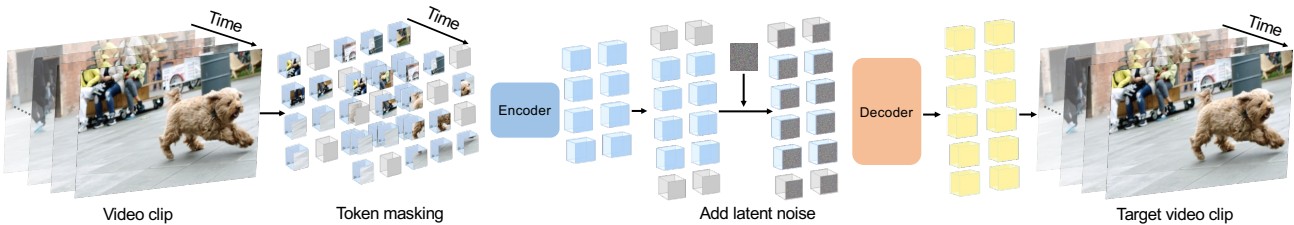

*Figure 2.* **An overview of our video tokenizer.** Our tokenizer is trained with masked-autoencoding and latent-denoising objectives (Section 3.3). The input video clip is first projected into a grid of spatio-temporal tokens, and a large portion of them are masked. The visible tokens are then processed by an encoder to produce latent representations. We add latent noise into latent feature to make the training consistent with downstream generative training. A decoder then reconstructs the original video clip from these corrupted latent tokens.

compact latent space (continuous or discrete), which makes training and inference feasible at scale. In this paper, we focus on diffusion-based video generators, and study how the tokenizer, which defines the data representation space of the diffusion process, affects downstream denoising-based learning and generation quality.

## 2.3. Masked Autoencoder

Masked autoencoders (MAEs) are a family of self-supervised methods for visual representation learning (He et al., 2022; Bao et al., 2022). Inspired by masked language modeling (e.g., BERT (Devlin et al., 2019)), MAEs reconstruct randomly masked patches from visible context using an encoder–decoder architecture. MAE-style learning has been successfully extended to videos (Tong et al., 2022; Wang et al., 2023; 2022) and other domains (Weinzaepfel et al., 2022; 2023; Pang et al., 2022; Zhang et al., 2023; Bachmann et al., 2022; Mizrahi et al., 2023).

While MAEs were originally developed for discriminative representation learning, recent work has begun adapting MAE-style training to tokenizers for diffusion models. In images, MAETok (Chen et al., 2025) uses masked modeling to obtain diffusion-friendly latents, and DeTok (Yang et al., 2025) further injects noise into latent tokens to better align tokenizer training with diffusion denoising. Our work is inspired by these directions but targets the video setting. We combine (i) MAE-style high-ratio masking with an efficient "encode-only-visible" encoder, and (ii) latent corruption via noise interpolation, to explicitly train a *corruption-inversion* video tokenizer whose latent space is robust under the corruptions encountered by downstream video diffusion models.

## 3. Method

We describe VideoMAETok, a video tokenizer designed for *latent video diffusion*. Our goal is to learn an autoencoding mapping that not only reconstructs videos, but also produces latent tokens that are *well-suited for denoising-based generative learning*.

## 3.1. Revisiting VideoMAE

VideoMAE (Tong et al., 2022) is a masked autoencoding framework for self-supervised video representation learning. It first partitions a video into a sequence of spatiotemporal patches (tokens), masks a large portion of tokens, and trains an encoder–decoder to reconstruct the missing content from visible tokens. A key practical advantage is efficiency: since the encoder processes only visible tokens, the computation scales with the number of unmasked tokens, enabling high masking ratios. VideoMAE typically uses a *heavy encoder* and a *light decoder*, as its primary goal is to learn discriminative representations for downstream understanding tasks.

## 3.2. Revisiting Variational Autoencoder

Variational Autoencoders (VAEs) (Kingma & Welling, 2013) learn a continuous latent representation with an explicit prior, typically $\mathcal{N}(\mathbf{0}, \mathbf{I})$. Compared with standard autoencoders (Bourlard & Kamp, 1988), VAEs regularize the encoder distribution $q(\mathbf{z}|\mathbf{x})$ to match the prior via a KL term (Kullback & Leibler, 1951), which encourages a smoother and more structured latent space. The model is commonly trained by *maximizing* the evidence lower bound (ELBO), or equivalently *minimizing* its negative form:

$$\mathcal{L}_{\text{VAE}} = \mathbb{E}_{q(\mathbf{z}|\mathbf{x})}\big[-\log p(\mathbf{x}|\mathbf{z})\big] + D_{\text{KL}}\big(q(\mathbf{z}|\mathbf{x}) \,\|\, p(\mathbf{z})\big), \quad (1)$$

where $\mathbf{x}$ denotes the input video (or its patch tokens) and $\mathbf{z}$ denotes the latent variables. This VAE-style regularization is widely used in continuous visual tokenizers (Rombach et al., 2022; Yao et al., 2025; Agarwal et al., 2025).

## 3.3. VideoMAETok

VideoMAETok is a ViT-based video tokenizer that combines (i) MAE-style masking and (ii) latent-space denoising, together with an asymmetric capacity allocation that favors the decoder. Figure 2 provides an overview of the architecture.

**Asymmetric autoencoder architecture.** Our tokenizer follows an encoder-decoder architecture based on the Vision Transformer (ViT), where input video frames are divided into non-overlapping patches, which are then linearly

projected into a sequence of embedding tokens. A key design choice (Figure 3) is an *asymmetric* architecture with a *lightweight encoder* and a *heavier decoder*, in contrast to VideoMAE which emphasizes a heavy encoder for discriminative representation learning. For generation-oriented tokenization, the encoder is mainly responsible for producing a compact latent representation, while the decoder must faithfully map latents back to visual tokens and RGB pixels. Insufficient decoder capacity can become the bottleneck that limits both reconstruction and the usefulness of latents for downstream diffusion training. We therefore allocate most parameters to the decoder, while keeping the encoder light for efficiency and scalability.

**Masked modeling.** We apply MAE-style random masking to the patch embeddings. Specifically, for each training batch, we determine the masking ratio as

$$m = \max\left(0, \mathcal{U}(-0.1, \rho)\right), \quad (2)$$

where $\rho$ denotes the maximum masking ratio. This sampling strategy exposes the tokenizer to a wide range of masking levels. Given the sampled ratio $m$, we randomly remove a subset of patch tokens before the encoder. Crucially, the encoder processes only the visible tokens, reducing the quadratic attention cost and enabling the use of high masking ratios in video settings. The decoder then reconstructs the complete set of visual tokens, as well as the final RGB video, from the latent representations of the visible tokens. This objective encourages the learned latent space to remain predictive even when substantial visual information is missing.

**Latent denoising.** Downstream diffusion and flow-matching generators learn to iteratively denoise corrupted latents (Ho et al., 2020; Rombach et al., 2022; Lipman et al., 2022; Ma et al., 2024). This process suggests that the latent space of a tokenizer should be robust to noise. To make the tokenizer training consistent with the subsequent training of the generative model, we integrate a denoising objective into our training pipeline, following the design in DeTok (Yang et al., 2025). This design operates by corrupting the latent embeddings through noise interpolation. For a given set of latent tokens $\mathbf{z}$ produced by the encoder, we interpolate them with Gaussian noise as follows:

$$\mathbf{z}' = (1 - \tau)\mathbf{z} + \tau\varepsilon(\gamma),$$
$$\text{where} \quad \varepsilon(\gamma) \sim \gamma \cdot \mathcal{N}(\mathbf{0}, \mathbf{I}), \quad \tau \sim \mathcal{U}(0, 1). \quad (3)$$

The latent $\mathbf{z}$ is corrupted by linearly interpolating it with noise $\varepsilon(\gamma)$. Here noise is sampled from a Gaussian distribution $\mathcal{N}(0, I)$, where $\gamma$ controls the noise scale and $\tau$ controls corruption intensity. Sampling $\tau$ over a continuous range encourages robustness across a spectrum of corruption magnitudes, rather than tuning to a single noise level.

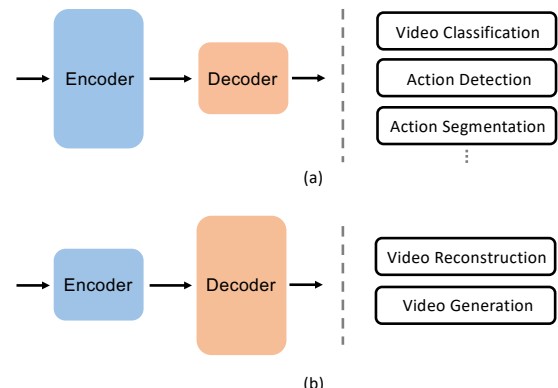

*Figure 3.* Encoder–decoder architecture design. (a) Original MAE-style design with a heavy encoder, optimized for downstream discriminative tasks (*e.g.* video classification). (b) VideoMAE-Tok uses a heavy decoder, which is more appropriate when the tokenizer is used to support video reconstruction and generation.

We apply this noise in the latent space to match the training paradigm of video diffusion models. During downstream latent diffusion training, we disable this extra corruption and encode videos normally (no masking and $\tau = 0$).

**Training objectives.** We train VideoMAETok with a weighted combination of standard losses for high-quality video synthesis, including pixel-level reconstruction, VAE-style KL regularization, perceptual loss, and an adversarial objective:

$$\mathcal{L} = \lambda_1 \mathcal{L}_{\text{MSE}} + \lambda_2 \mathcal{L}_{\text{KL}} + \lambda_3 \mathcal{L}_{\text{percep}} + \lambda_4 \mathcal{L}_{\text{GAN}}, \quad (4)$$

where each $\lambda$ is a hyperparameter that controls the impact of each loss term in the overall objective. The key difference from reconstruction-oriented tokenizers is not the loss components themselves, but *the training condition*: the decoder is optimized to reconstruct clean videos from *strongly corrupted* latent tokens (masking + interpolative noise), thereby shaping a latent space that is more robust and better aligned with downstream denoising-based generation.

## 4. Experiments

### 4.1. Datasets and Evaluation

We use three common video datasets: Kinetics-400 (Kay et al., 2017), Kinetics-600 (Carreira et al., 2018), and UCF-101 (Soomro et al., 2012) for training and evaluation. The Kinetics-400 (Kay et al., 2017) dataset is composed of approximately 240,000 training videos and 20,000 validation videos from 400 classes, with each clip lasting about 10 seconds. Kinetics-600 (Carreira et al., 2018) is an extension of Kinetics-400, containing approximately 390,000 training videos and 30,000 validation videos across 600 action categories. UCF-101 (Soomro et al., 2012) is a smaller video dataset, with 13,320 videos from 101 action categories.

| Size | Hidden Size | Blocks | Heads | Param. |
|------|-------------|--------|-------|--------|
| Small (S) | 512 | 8 | 8 | 25.8M |
| Base (B) | 768 | 12 | 12 | 85.9M |
| Large (L) | 1024 | 24 | 16 | 307M |

*Table 1.* **Tokenizer configurations.** We summarize the architectural settings and parameter counts of the encoder and decoder.

We first train VideoMAETok on either the Kinetics-400 or Kinetics-600 dataset. To verify the tokenizer's effectiveness for latent diffusion models, we then conduct class-conditional video generation experiments on UCF-101 and Kinetics-600. Specifically, we train a latent diffusion model (SiT (Ma et al., 2024) or LightningDiT (Yao et al., 2025)) on top of the latent features extracted by our VideoMAE-Tok. We use reconstruction Fréchet Video Distance (rFVD) to evaluate reconstruction quality and generation Fréchet Video Distance (gFVD) to assess generation quality.

### 4.2. Implementation Details

**Video tokenizer model.** We provide the detailed model parameters of our tokenizers in Table 1. For both the encoder and decoder, we implement our tokenizer using the plain Vision Transformer (ViT) architecture, with several small changes inspired by LLaMA (Touvron et al., 2023), including Rotary Positional Embeddings (RoPE) (Su et al., 2024), RMSNorm (Zhang & Sennrich, 2019) for pre-normalization, and the SwiGLU (Shazeer, 2020) activation function. Following standard practice, we use learned positional embeddings in conjunction with RoPE. By default, the input video resolution is set to $16 \times 256 \times 256$. We employ a patch embedding with $2 \times 16 \times 16$, which transforms the input video into a sequence of 2048 tokens. The latent dimension of our tokenizers is set to 16.

**Video tokenizer training.** For our ablation studies, we train a model composed of a ViT-S encoder and a ViT-B decoder for 200 epochs on the Kinetics-400 dataset, with the GAN loss introduced at epoch 100. For our final experiments, we use the Kinetics-600 dataset to train tokenizers with three architectural variants, corresponding to Small-Base (S-B), Base-Base (B-B), and Base-Large (B-L) encoder-decoder configurations. In both training settings, we use a total batch size of 512. The learning rate schedule consists of a linear warm-up to a peak value of $4.0 \times 10^{-4}$, which is then followed by a cosine decay. We use the AdamW optimizer with $\beta$ parameters set to (0.9, 0.95) and a weight decay of $1.0 \times 10^{-4}$. We use 64 A100 GPUs for model training. The training of the VideoMAETok (S-B) model on Kinetics-400 for 200 epochs requires approximately 22 hours. On Kinetics-600, the training times for the S-B, B-B, and B-L model variants are approximately 34 hours, 39 hours, and 54 hours, respectively.

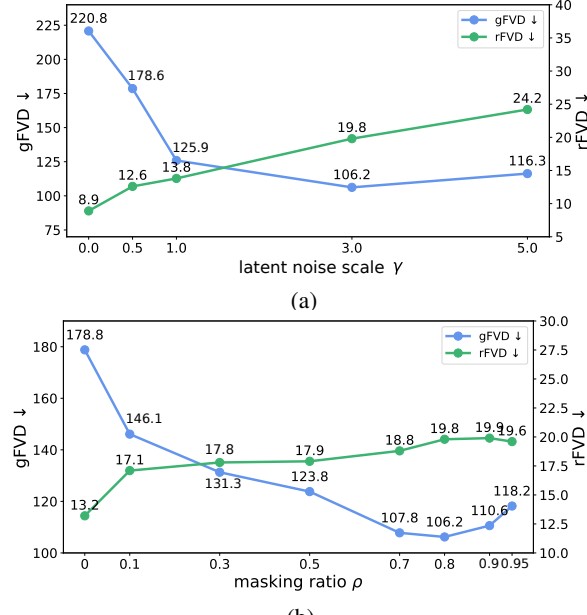

(a)

(b)

*Figure 4.* The effect of **noise scale** and **masking ratio**.

**Generation model and training.** For ablation studies, we use the SiT-B architecture (Ma et al., 2024) and train the model for 100k iterations, applying an exponential moving average (EMA) with a decay rate of 0.999. For our final experiments, we employ LightningDiT-XL (Yao et al., 2025) and SiT-XL (Ma et al., 2024) for better performance and train each model for 300k iterations with an EMA decay of 0.9999. We adopt the default samplers from the original implementations and use 250 denoising steps during the sampling phase. For all experiments reported in this paper, we follow the training recipe: a global batch size of 256, the AdamW (Loshchilov & Hutter, 2017) optimizer, and a constant learning rate of $1 \times 10^{-4}$.

### 4.3. Ablation Studies

We conduct a series of ablation studies to analyze key design choices of our method. We evaluate performance under two different metrics: reconstruction and generation. For reconstruction, we report rFVD on the Kinetics-400 validation set. For generation, we train a SiT-Base generator on UCF-101 for 100,000 iterations and report the gFVD with a Classifier-Free Guidance (CFG) scale of 1.8.

**Noise scale.** We conduct an ablation study to analyze the effect of noise scale $\gamma$ as in Equation 3. We vary $\gamma$ across the values $\{0.0, 0.5, 1.0, 3.0, 5.0\}$. As shown in Figure 4 (a), the results indicate that while rFVD worsens as $\gamma$ increases, gFVD shows an opposite trend, improving significantly and reaching its minimum value at $\gamma = 3.0$. A further increase in $\gamma$ to 5.0 leads to a degradation in gFVD. This suggests that introducing an appropriate amount of noise effectively

| latent noise? | masking? | rFVD ↓ | gFVD ↓ |
|:---:|:---:|:---:|:---:|
| ✗ | ✗ | **6.0** | 418.8 |
| ✗ | ✓ | 8.9 | 220.8 |
| ✓ | ✗ | 13.2 | 178.8 |
| ✓ | ✓ | 19.8 | **106.2** |

*(a)* **Components analysis.** Corruption-based tokenizer training is crucial for downstream diffusion-based generation. The first row corresponds to VAE baseline.

| tokenizer | rFVD ↓ | gFVD ↓ |
|:---:|:---:|:---:|
| ViT S-S | 27.6 | 172.9 |
| ViT S-B | 19.8 | 106.2 |
| ViT B-B | 18.0 | 94.2 |
| ViT B-L | **16.4** | **87.7** |

*(b)* **Scalability of tokenizer.** Larger tokenizers consistently improve reconstruction and generation.

| generator | param. | gFVD ↓ |
|:---:|:---:|:---:|
| SiT-S | 33M | 237.9 |
| SiT-B | 131M | 106.2 |
| SiT-L | 458M | 85.6 |
| SiT-XL | 675M | **82.2** |

*(c)* **Scalability of generator.** Larger generators lead to better generation quality.

| $\lambda_1$ | rFVD ↓ | gFVD ↓ |
|:---:|:---:|:---:|
| ×1 | 19.8 | **106.2** |
| ×2 | 19.1 | 141.9 |
| ×5 | 18.2 | 171.7 |
| ×10 | **9.5** | 3998.2 |

*(d)* **Reconstruction weight** $\lambda_1$. Prioritizing reconstruction harms generation performance.

| #channel | 4 | 8 | 16 | 32 |
|:---:|:---:|:---:|:---:|:---:|
| rFVD ↓ | 25.1 | 20.3 | **19.8** | 21.1 |
| gFVD ↓ | 1427 | 336 | **106.2** | 3948 |

*(e)* **Number of Latent channels.**

| | rFVD ↓ | gFVD ↓ |
|:---|:---:|:---:|
| Non-Causal Attention | **19.8** | **106.2** |
| Causal Attention | 24.5 | 125.4 |

*(f)* **Causal vs. Non-Causal.**

*Table 2.* **Ablation studies and scalability analysis.** All tokenizers take 16-frame inputs and are pre-trained on the Kinetics-400 training set for 200 epochs. For video reconstruction, we report rFVD on the Kinetics-400 validation set. For class-conditional video generation, we train a SiT (Ma et al., 2024) generator on UCF-101 for 100k iterations, and report gFVD as the metric. The default tokenizer is ViT S-B, pretrained with a maximum masking ratio $\rho = 80\%$ and $\gamma = 3.0$, which is highlighted in gray.

aligns the tokenizer's training with the objectives of the video generator. Based on these results, we use $\gamma = 3.0$ as our default setting for all subsequent experiments.

**Masking ratio.** We investigate the impact of the maximum masking ratio $\rho$ on both reconstruction and generation performance. We vary $\rho$ across the values $\{0, 0.1, 0.3, 0.5, 0.7, 0.8, 0.9, 0.95\}$. As shown in Figure 4 (b), the reconstruction FVD (rFVD) is best with no masking ($\rho = 0$) and degrades as the masking ratio increases. This is expected, as a higher masking ratio presents a more difficult reconstruction challenge. Conversely, the generation FVD (gFVD), which is the more critical metric for our objective, shows a significant improvement with a higher masking ratio. The gFVD score consistently improves as $\rho$ increases, reaching its best value of 106.2 at $\rho = 0.8$, before degrading at $\rho = 0.9$. This demonstrates that a high masking ratio encourages the tokenizer to learn more semantically meaningful and robust representations, which are highly beneficial for the downstream video generation task. Since our primary goal is generative modeling, we prioritize the gFVD metric and select $\rho = 0.8$ as our default setting.

**Components analysis.** We ablate the two types of corruption discussed above in Table 2a. Without either corruption, the tokenizer reduces to a VAE-style baseline with best reconstruction (6.0 rFVD) but very poor generative quality (418.8 gFVD), as it only learns to invert clean inputs and lacks the capability to reconstruct under corruption. Adding masking or latent noise alone slightly worsens rFVD but substantially improves gFVD, indicating a better alignment with the downstream diffusion model. Combining both yields the best generative performance (106.2 gFVD). Qualitative results in Fig. 5 further show that this setting produces latents that can survive strong corruption and still reconstruct coherent videos, with masking and latent noise acting as

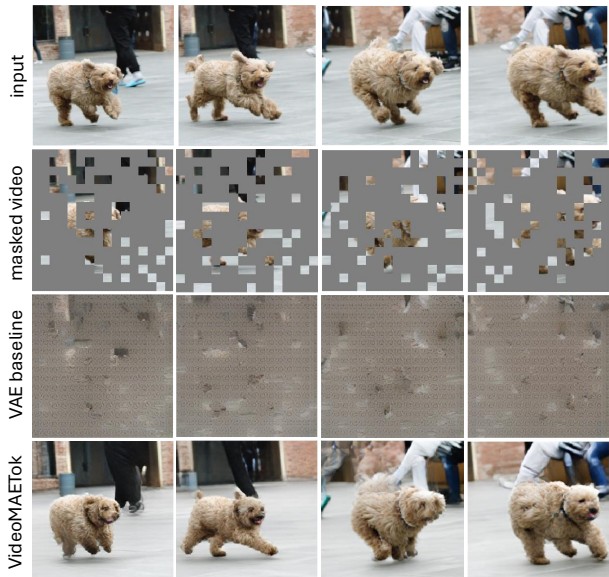

*Figure 5.* Qualitative comparison of reconstructions from heavily corrupted inputs using the VAE baseline and VideoMAETok (masking ratio $\rho = 0.8$, latent noise scale $\gamma = 3.0$). We uniformly sample 4 frames from each 16-frame clip for visualization.

complementary corruption mechanisms.

To better understand how our design reshapes the latent space, we further visualize the token embeddings using t-SNE on 10,000 videos from Kinetics-400, as shown in Fig. 1. Compared to the VAE baseline, whose latents form entangled and partially collapsed clusters, VideoMAETok yields a more structured and well-spread latent manifold, with smoother transitions and clearer separation between regions. This corroborates our quantitative findings and suggests that corruption-aware training not only improves gFVD, but also induces a better latent space.

| Method | Training Data | #Tokens | #Dim. | TFLOPs | Generator | Kinetics-600 | | UCF-101 | |
|---|---|---|---|---|---|---|---|---|---|
| | | | | | | rFVD↓ | gFVD↓ | rFVD↓ | gFVD↓ |
| OmniTokenizer (Wang et al., 2024) | IN-1K+K600 | 4096 | 8 | 5.82 | LightningDiT-XL | 5.14 | 242.54 | 10.21 | 88.89 |
| Cosmos-S (Agarwal et al., 2025) | proprietary data | 512 | 16 | 0.49 | | 70.26 | 210.21 | 104.51 | 191.49 |
| Cosmos-M (Agarwal et al., 2025) | proprietary data | 2048 | 16 | 2.37 | | 6.77 | 125.02 | 13.67 | 85.22 |
| Cosmos-L (Agarwal et al., 2025) | proprietary data | 4096 | 16 | 5.82 | | **3.28** | 302.58 | **5.55** | 75.11 |
| LTX (HaCohen et al., 2024) | proprietary data | 128 | 128 | 0.12 | | 22.11 | 358.48 | 35.32 | 345.82 |
| LARP-L (Wang et al., 2025) | UCF+K600 | 1024 | 16 | - | | 23 | - | 35 | 99 |
| VFRTok-S (Zhong et al., 2025) | IN-1K+K600 | 128 | 128 | 0.12 | | 6.02 | 131.34 | 15.55 | 129.55 |
| VFRTok-L (Zhong et al., 2025) | IN-1K+K600 | 512 | 32 | 0.49 | | 4.64 | 124.78 | 13.79 | 71.34 |
| VideoMAETok (S-B) | K600 | 2048 | 16 | 0.23 | SiT-XL | 16.12 | 58.54 | 31.96 | 65.43 |
| VideoMAETok (B-L) | K600 | 2048 | 16 | 0.78 | | 10.01 | 47.76 | 21.78 | 52.54 |
| VideoMAETok (S-B) | K600 | 2048 | 16 | 0.23 | LightningDiT-XL | 16.12 | 52.43 | 31.96 | 61.89 |
| VideoMAETok (B-L) | K600 | 2048 | 16 | 0.78 | | 10.01 | **42.16** | 21.78 | **49.93** |

*Table 3.* Comparison of class-conditional video generation on Kinetics-600 and UCF-101. We report both reconstruction FVD (rFVD↓) and generation FVD (gFVD↓). Overall best gFVD values are **bolded**.

**Reconstruction vs. generation.** We examine the relationship between the tokenizer's reconstruction fidelity and the performance of the final generative model. The results are shown in Table 2d. To do this, we increase the weight of the reconstruction loss, $\lambda_1$ in Eq. 4, during the tokenizer's training. While this modification significantly improves the reconstruction performance, with the rFVD dropping from 19.8 to 9.5, the resulting feature space leads to a failure in training the generation model. The gFVD score deteriorates to a very poor value of 3998. This result highlights that the reconstruction task and its corresponding metric, rFVD, are not indicative of the final generative model's performance. An excessive focus on reconstruction quality can, in fact, be detrimental to the training of the ultimate generative model.

**Number of latent channels.** Table 2e shows that increasing the latent width from 4 to 16 channels consistently improves both reconstruction (rFVD from 25.1 to 19.8) and generation quality (gFVD from 1427 to 106.2), suggesting that the 4-channel design of SD-VAE (8×8 spatial compression) may be insufficient in our video generation setting. In contrast, further enlarging the width to 32 channels yields worse rFVD and severely degraded gFVD (3948), indicating that overly high-dimensional latents make diffusion training harder to optimize (Yao et al., 2025). Overall, 16 channels strike a good balance between sufficient capacity for video and stable generative training.

**Causal vs. non-causal.** Given the streaming nature of video, we also consider replacing full self-attention with frame-level causal attention. As shown in Table 2f, causal attention consistently hurts both reconstruction (rFVD from 19.8 to 24.5) and generation quality (gFVD from 106.2 to 125.4). We hypothesize that the causal mask restricts token interactions across time, which is undesirable for a tokenizer whose primary goal is to aggressively compress videos while exploiting rich bidirectional spatiotemporal

context. In contrast, non-causal attention allows free information flow among all frames, leading to a more expressive latent space and better generation quality.

### 4.4. Scalability

We investigate the scalability of our approach by independently increasing the size of the tokenizer and the generative model. First, as shown in Table 2b, we fix the generator as SiT-Base and scale the tokenizer from ViT S-S to ViT S-B, ViT B-B, and then to ViT B-L. The results demonstrate a consistent improvement in both reconstruction FVD (rFVD) and generation FVD (gFVD) as the tokenizer's capacity increases. This indicates that a more powerful tokenizer learns more effective representations, which benefits both the reconstruction and the downstream generation task. Next, we fix the tokenizer to our default ViT S-B model and scale the generator from SiT-Base to SiT-Large and SiT-XL. As detailed in Table 2c, increasing the size of the generator leads to a substantial improvement in gFVD, from 106.2 to 82.2. These experiments collectively show that both the tokenizer and the generator are scalable, and increasing the capacity of either part leads to better video generation performance.

### 4.5. Comparison with the state-of-the-art

In this section, we compare our VideoMAETok with several recent state-of-the-art tokenizers, including OmniTokenizer (Wang et al., 2024), Cosmos (Agarwal et al., 2025), LTX (HaCohen et al., 2024), and VFRTok (Zhong et al., 2025). We first train a family of VideoMAETok tokenizers on Kinetics-600, and then pair them with two diffusion-based video generators: SiT-XL and LightningDiT-XL. For each tokenizer, we freeze the tokenizer and train the generator on top of the latent features. Table 3 summarizes the results in terms of rFVD and gFVD. While some tokenizers, such as Cosmos-L, achieve very low rFVD, indicating

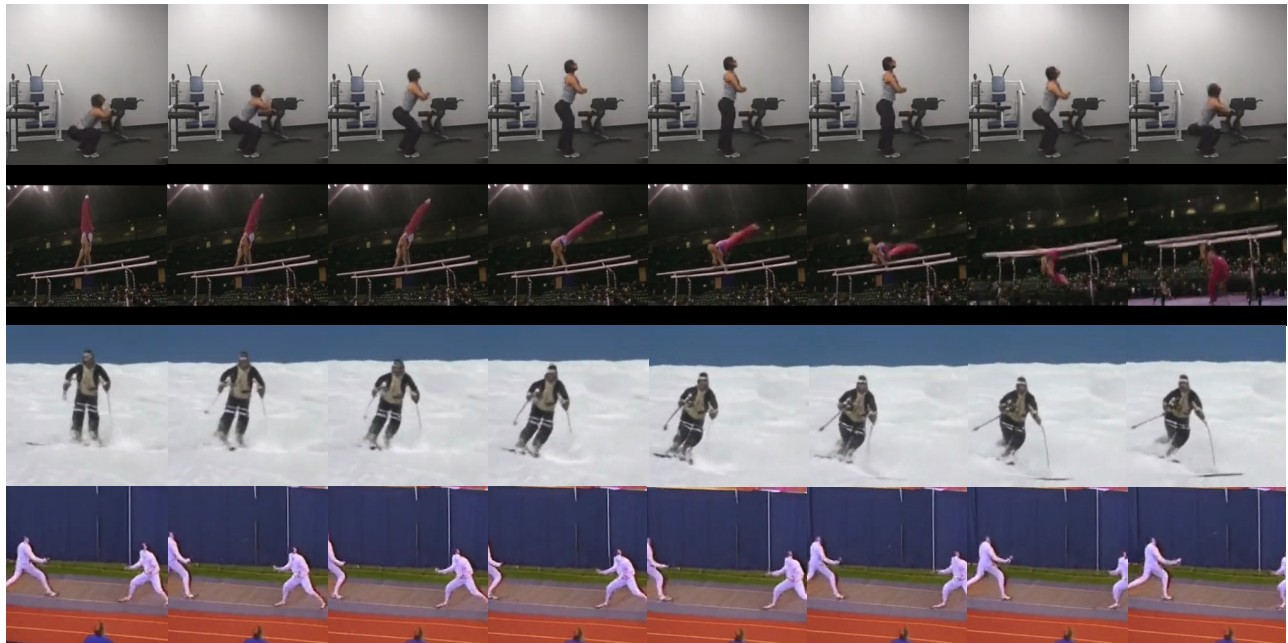

*Figure 6.* **Class-conditional video generation samples on UCF-101.** All samples are produced with VideoMAETok-B-L and SiT-XL as the diffusion generator. For clearer visualization, we uniformly sample 8 frames from each generated 16-frame video sequence. We also provide some reconstruction examples in Figure 7.

strong reconstruction fidelity, their gFVD scores are significantly higher. This trend is consistent with our ablation studies and supports the observation that optimizing a tokenizer solely for pixel-level reconstruction does not necessarily yield a better latent space for generative modeling.

In contrast, VideoMAETok achieves a trade-off between reconstruction quality and generative performance. With SiT-XL as the generator, our largest tokenizer, VideoMAETok (B-L), reaches gFVD scores of 47.76 on Kinetics-600 and 52.54 on UCF-101, already establishing a new state of the art among the compared tokenizers. When switching to LightningDiT-XL, VideoMAETok (B-L) further improves the generation performance, achieving gFVD scores of 42.16 on Kinetics-600 and 49.93 on UCF-101. These results substantially outperform all prior tokenizers; for example, the best competing baseline, VFRTok-L, attains gFVD scores of 124.78 on Kinetics-600 and 71.34 on UCF-101.

Our models are also computationally efficient. Despite using a comparable or smaller number of tokens and latent dimensions, the largest VideoMAETok variant (B-L) requires only 0.78 TFLOPs, which is an order of magnitude more efficient than Cosmos-L (5.82 TFLOPs). Even when compared to the most recent VFRTok family, our tokenizers consistently deliver better gFVD scores at similar or lower computational cost. These results demonstrate that VideoMAETok learns latent representations that are particularly well suited to high-quality video generation, making it a practical tokenizer for video generation.

## 4.6. Qualitative Results

We provide qualitative results of our class-conditional video generation on UCF-101 in Figure 6. Each row shows one video sequence (covering roughly 2 seconds) generated by our diffusion model conditioned on the target action label. Despite the relatively short temporal window, the model is able to capture the characteristic motion patterns and scene layouts of diverse actions, such as squatting in a gym environment and skiing downhill. The generated video clips exhibit coherent temporal dynamics: human poses evolve smoothly over time, object and background appearances remain consistent across frames, and action-specific details (e.g., arm and leg articulation, body orientation changes, and interaction with scene elements) are preserved.

## 5. Conclusion

In this work, we presented VideoMAETok, a family of Transformer-based video tokenizers explicitly designed for latent video diffusion models. By training masked autoencoders to reconstruct clean videos from heavily corrupted latents via random masking and latent noise, VideoMAETok learns robust representations that are better suited to the denoising process of diffusion models than those produced by reconstruction-oriented VAEs. We hope VideoMAETok can serve as a practical tokenizer for future research on high-fidelity, controllable video generation.

## Impact Statement

This work improves the efficiency and quality of diffusion-based video generation by introducing new video tokenizers. Potential positive impacts include enabling better video representation and higher-quality video content creation. However, stronger video generation models may also increase the risk of misuse, such as generating misleading or non-consensual synthetic videos. We encourage future work to pair advances in video generation with safeguards, including provenance, dataset curation, and responsible deployment.

## Acknowledgements

This project has received funding from the European Research Council (ERC) under the European Union's Horizon 2020 research and innovation programme (grant agreement n° 101021347). This project also received funding from the Flemish Government under the "Onderzoeksprogramma Artificiële Intelligentie (AI) Vlaanderen" and the Research Foundation Flanders (FWO) through project number G058826N. We acknowledge EuroHPC JU for awarding the project IDs EHPC-AI-2024A03-029 and EHPC-AI-2024A06-035 access to the EuroHPC supercomputer LEONARDO, hosted by CINECA (Italy) and the LEONARDO consortium.

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

In this appendix, we present additional video reconstruction results and visualizations in Section A, text-to-video generation experiments in Section B, temporal-window analyses in Section C, and a discussion of closely related work and limitations in Section D.

## A. Video Reconstruction

For completeness, we evaluate reconstruction quality using metrics including Peak Signal-to-Noise Ratio (PSNR), Structural Similarity Index Measure (SSIM) (Wang et al., 2004), Learned Perceptual Image Patch Similarity (LPIPS) (Zhang et al., 2018), and reconstruction Fréchet Video Distance (rFVD) (Unterthiner et al., 2018). The results on Kinetics-600 (Carreira et al., 2018) and UCF-101 (Soomro et al., 2012) are shown in Table 6. Since our tokenizer is trained to invert corrupted latents through random masking and latent noise, it is not explicitly optimized for pixel-level fidelity. The reconstruction metrics of VideoMAETok are initially worse than other tokenizers such as Cosmos (Agarwal et al., 2025) or VFRTok (Zhong et al., 2025). We further freeze the encoder and perform an additional 50 epochs of decoder fine-tuning on *clean* latent embeddings (marked with †). This simple fine-tuning stage can boost reconstruction fidelity across all metrics. On Kinetics-600, PSNR improves from 28.69 to 30.96 for VideoMAETok S-B and from 26.72 to 31.23 for VideoMAETok B-L, accompanied by consistent gains in SSIM and LPIPS, and a reduction in rFVD (e.g., from 16.12 to 12.18 for VideoMAETok S-B). Similar trends are observed on UCF-101. After decoder fine-tuning, our tokenizer achieves reconstruction quality that is competitive with prior work, despite being trained only on Kinetics-600. Notably, most competing tokenizers are trained on ImageNet-1K (Russakovsky et al., 2015) or related clean image corpora, whereas Kinetics consists of YouTube videos that often contain motion blur, compression artifacts, and generally lower visual quality. We expect that performing decoder fine-tuning on higher-quality video datasets would further improve reconstruction fidelity, while still preserving the learned latent space, at only modest additional computational cost. We also show some qualitative examples in Figure 7. The reconstructed videos closely track the original ones in terms of global structure, motion dynamics, and object appearance, with visible discrepancies largely confined to textures and background clutter.

## B. Text-to-Video Generation

We further evaluate VideoMAETok in a text-to-video generation setting. Specifically, we compare SD-VAE, ViT-VAE, and VideoMAETok as video tokenizers while keeping the generation model and training setup fixed. For each tokenizer, we train an MMDiT (Esser et al., 2024) model on

| SD-VAE | VAE (Baseline) | VideoMAETok |
|--------|----------------|-------------|
| 74.56  | 69.78          | **78.82**   |

Table 4. Text-to-video generation comparison on VBench.

| Frames | rFVD ↓ | gFVD ↓ |
|--------|--------|--------|
| 8      | 22.3   | 103.7  |
| 16     | 19.8   | 92.2   |
| 32     | **18.4** | **86.4** |

Table 5. Effect of the tokenizer temporal window.

OpenVid-1M (Nan et al., 2025) for 40k iterations with an exponential moving average (EMA). The models generate 64-frame videos with a temporal stride of 2, corresponding to approximately 4 seconds of video. We evaluate the generated videos using VBench (Huang et al., 2024). As shown in Table 4, VideoMAETok achieves the best VBench total score, outperforming both SD-VAE and ViT-VAE under the same setting.

## C. Effect of Temporal Window

Our main experiments use 16-frame clips to maintain a controlled comparison and remain consistent with the tokenizer and generator configurations. Nevertheless, VideoMAETok is not conceptually restricted to a fixed temporal window. To study its ability to model longer videos, we additionally train tokenizers with temporal windows of 8, 16, and 32 frames under the same ablation setting on the Kinetics-400 training set for 200 epochs. For a fair comparison of generation quality, we train a 32-frame SiT on top of each tokenizer. For the 8-frame and 16-frame tokenizers, we encode and decode 32-frame videos in a chunk-wise manner.

As shown in Table 5, increasing the temporal window of the tokenizer consistently improves both reconstruction and generation quality. In particular, extending the window from 16 to 32 frames reduces rFVD from 19.8 to 18.4 and gFVD from 92.2 to 86.4. These results suggest that VideoMAETok can effectively benefit from longer temporal contexts, providing a promising direction for scaling the tokenizer to longer video generation.

## D. Discussion

Several recent works are closely related to ours, most notably MAETok (Chen et al., 2025) and ViTok (Hansen-Estruch et al., 2025), which also develop continuous visual tokenizers for diffusion models. In what follows, we discuss these methods in detail and highlight how VideoMAETok differs from them and what contributions it makes.

**Relation to MAETok.** MAETok (Chen et al., 2025) is designed primarily as an *image* tokenizer for latent diffu-

| Method | Training Data | #Dim. | Kinetics-600 | | | | UCF-101 | | | |
|---|---|---|---|---|---|---|---|---|---|---|
| | | | PSNR↑ | SSIM↑ | LPIPS↓ | rFVD↓ | PSNR↑ | SSIM↑ | LPIPS↓ | rFVD↓ |
| OmniTokenizer (Wang et al., 2024) | IN-1K+K600 | 8 | 29.35 | 0.9143 | 0.0573 | 5.14 | 28.95 | 0.9239 | 0.0505 | 10.21 |
| Cosmos-L (Agarwal et al., 2025) | proprietary data | 16 | 33.34 | 0.9284 | 0.0546 | 3.28 | 33.42 | 0.9372 | 0.0439 | 5.55 |
| Cosmos-M (Agarwal et al., 2025) | proprietary data | 16 | 31.66 | 0.9068 | 0.0710 | 6.77 | 31.70 | 0.9177 | 0.0575 | 13.67 |
| Cosmos-S (Agarwal et al., 2025) | proprietary data | 16 | 28.46 | 0.8445 | 0.1209 | 70.26 | 28.26 | 0.8577 | 0.1046 | 104.51 |
| LTX (HaCohen et al., 2024) | proprietary data | 128 | 32.04 | 0.9100 | 0.0582 | 22.11 | 32.02 | 0.9202 | 0.0508 | 35.32 |
| VFRTok-S (Zhong et al., 2025) | IN-1K+K600 | 128 | 31.55 | 0.9089 | 0.0401 | 6.02 | 31.50 | 0.9178 | 0.0401 | 15.55 |
| VFRTok-L (Zhong et al., 2025) | IN-1K+K600 | 32 | 31.63 | 0.9104 | 0.0394 | 4.64 | 31.54 | 0.9193 | 0.0391 | 13.79 |
| VideoMAETok (S-B) | K600 | 16 | 28.69 | 0.8186 | 0.0948 | 16.12 | 26.68 | 0.8181 | 0.0653 | 31.96 |
| VideoMAETok (B-L) | K600 | 16 | 26.72 | 0.8643 | 0.0876 | 10.01 | 28.36 | 0.8674 | 0.0582 | 21.78 |
| VideoMAETok (S-B)† | K600 | 16 | 30.96 | 0.8836 | 0.0645 | 12.18 | 30.23 | 0.8743 | 0.0513 | 23.72 |
| VideoMAETok (B-L)† | K600 | 16 | 31.23 | 0.9023 | 0.0504 | 8.64 | 31.76 | 0.9016 | 0.0423 | 17.95 |

*Table 6.* Comparison of video reconstruction performance on Kinetics-600 and UCF-101. After decoder fine-tuning, our tokenizer achieves reconstruction quality that is competitive with prior work, despite being trained only on Kinetics-600. † denotes additional decoder fine-tuning.

sion models. It uses a ViT-based autoencoder trained with masked modeling to obtain a more discriminative latent space, and is coupled with DiT (Peebles & Xie, 2023) for image generation. In contrast, our VideoMAETok is explicitly a *video* tokenizer. Our encoder and decoder operate on spatio-temporal tokens and are optimized for video diffusion generation on benchmarks such as Kinetics-600 and UCF-101. Moreover, MAETok enriches its latent space via auxiliary decoders that regress HOG, DINOv2 (Oquab et al., 2023), and CLIP (Radford et al., 2021) features for masked tokens, effectively performing semantic distillation from strong external teachers. Our VideoMAETok deliberately avoids such auxiliary heads and relies solely on reconstructing clean videos from heavily corrupted latents. Additionally, MAETok keeps masked tokens in the sequence and predicts their features, whereas we discard masked visual tokens in the encoder as in original MAEs (He et al., 2022; Tong et al., 2022), which reduces training cost. Our experiments show that this is sufficient to obtain a structured latent space and competitive generation performance, even when trained solely on video data.

**Relation to ViTok.** ViTok (Hansen-Estruch et al., 2025) adopts a plain ViT as the backbone of a continuous visual tokenizer and systematically studies how scaling encoder/decoder capacity impacts reconstruction and generation for both images and videos. However, ViTok follows a conventional VAE-style training recipe, using a two-stage objective that combines MSE, LPIPS, KL, and GAN losses, and is optimized primarily on clean RGB pixels. This leads to good reconstruction results but the relationship between reconstruction-oriented scaling and downstream generation quality is non-trivial, with larger bottlenecks sometimes hurting generative performance. Our VideoMAETok instead starts from the opposite end of this trade-off. We optimize an autoencoder directly under strong corruption (random masking and latent noise) to obtain a latent space that is better aligned with the diffusion denoising process.

Finally, ViTok is trained on extremely large-scale, curated image and video corpora such as 450M Shutterstock images and 30M Shutterstock videos, whereas our work focuses on a much smaller but widely used video dataset (*e.g.* Kinetics-600 with only 390K video clips). Our results thus show that VideoMAETok can already yield strong generation performance in a smaller data regime.

**Limitations and Future Work.** The introduction of corruption during tokenizer training inevitably degrades reconstruction quality (rFVD). Although fine-tuning the decoder on clean latent embeddings partially mitigates this discrepancy, developing training schemes that better balance reconstruction and generation objectives is an open challenge. Our current design does not explicitly enforce high-level semantic alignment in the latent space. Incorporating semantic distillation (Yao et al., 2025; Chen et al., 2025) is a promising direction to further enhance the semantic representation of the learned tokens.

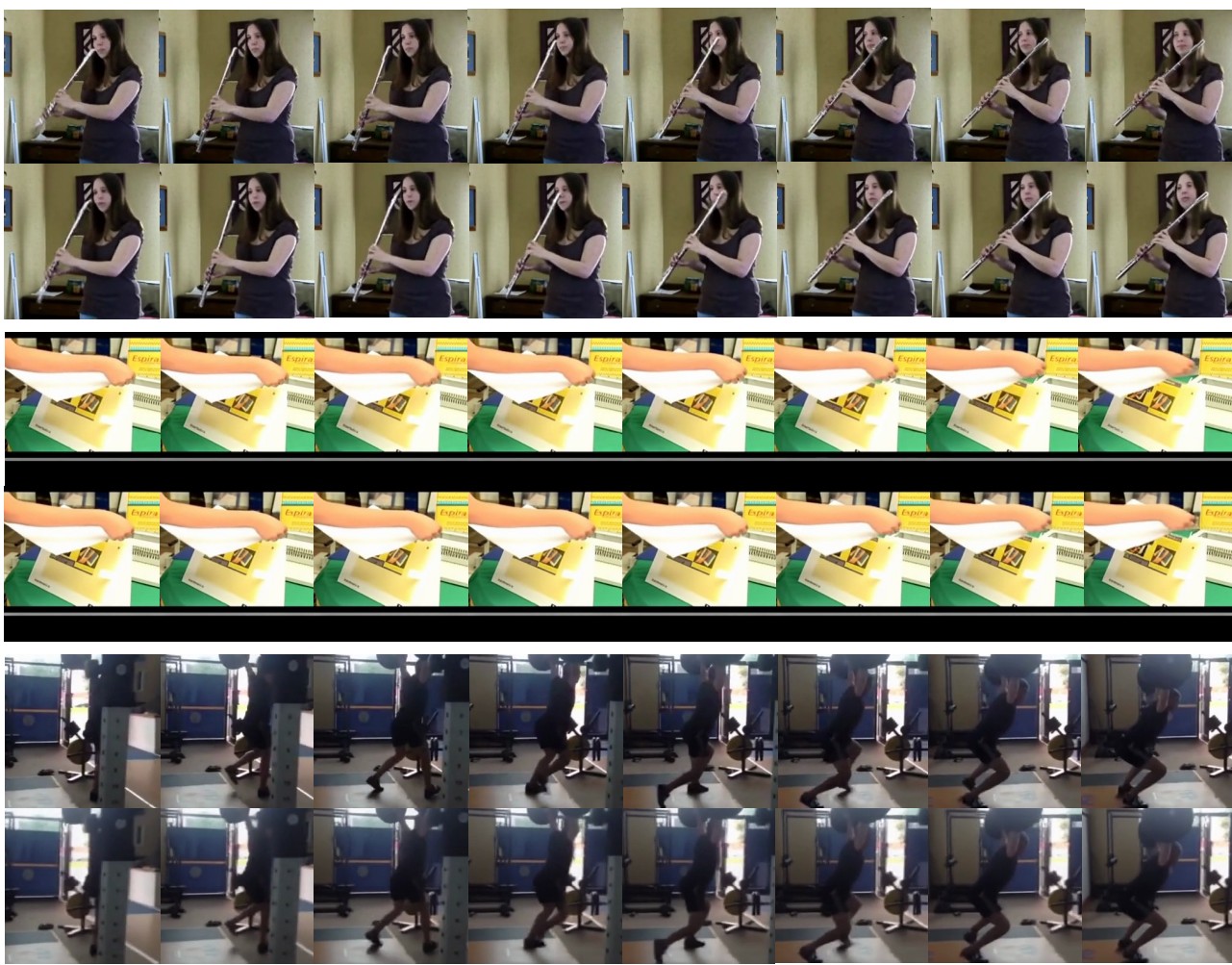

*Figure 7.* **Qualitative video reconstruction results.** We show three example clips from Kinetics-600, the top row in each pair is the original video and the bottom row is the reconstructed video produced by our VideoMAETok B-L. For clearer visual comparison, we uniformly sample 8 frames from each video sequence.

