# OpenReview forum: "VideoMAETok: Boosting Video Diffusion Models via Masked Autoencoders as Tokenizers"
_ICML.cc/2026/Conference — ICML 2026 regular_

### Official Review · Reviewer_Qpw8 · 2026-03-05

**Soundness:** 3
**Presentation:** 3
**Significance:** 3
**Originality:** 2
**Overall Recommendation:** 4
**Confidence:** 4

**Summary:**

This paper proposes training a video tokenizer using the MAE objective. Unlike MAE, it uses a small encoder and a large decoder. Specifically, the method (1) applies high-ratio token masking to the encoder input, similar to MAE, and (2) injects noise into the latent representations to better align with the downstream denoising process. The experimental results show that the method improves generation quality while achieving reasonable reconstruction quality.

**Compliance With Llm Reviewing Policy:**

Affirmed.

**Final Justification:**

The rebuttal addressed my concerns so I raise my score.

**Key Questions For Authors:**

Questions / Weakness:

(1) How is the plot in Figure 1 generated? Does a more uniformly distributed latent distribution necessarily indicate a better representation than another one? Also, there seems to be no discussion of Figure 1 in the main text.

(2) It seems that the large mask ratio could be a fundamental bottleneck for reconstruction, which might limit follow-up work. Do the authors have any ideas on how to improve this? For example, would mixing training data with mask ratios of 0 and 0.8 help? A 0 mask ratio does not seem to introduce much overhead given the lightweight encoder.

(3) This is not my major concern, but I feel the paper somewhat lacks novelty because these ideas have already been studied in the image domain. I understand that applying and validating these design choices in the video domain is itself a contribution, but it may not be sufficient.

(4) Can the authors provide a computation and memory cost comparison between the proposed architecture and video tokenizer baselines, as well as VideoMAE?

**Limitations:**

yes

**Strengths And Weaknesses:**

Strengths:
(1) The architecture is simple and easy to implement.
(2) While masked autoencoder–based tokenizer training has been explored in the image domain, it is less studied for video generation. This work provides useful insights into designing semantic tokenizers for the video domain.
(3) The paper is generally well written and easy to follow.
(4) The experiments are relatively comprehensive and include multiple ablations that help validate the proposed design choices and empirical claims.

Weaknesses:
Please see the questions below.

---

> ### Author Rebuttal · Authors · 2026-03-31
>
> We thank the reviewer Qpw8 for the detailed comments and for recognizing the simplicity of the architecture, the clarity of the presentation, and the breadth of the ablations.
>
> **(1) Figure 1: how is it generated, and what does it mean?**
>
> The embeddings in Figure 1 were extracted from a frozen tokenizer on 10,000 videos from the Kinetics-400 validation set, as described in Sec. 4.3 (L307–316). We agree that the t-SNE should be interpreted as qualitative intuition, not as a standalone proof. A more uniformly distributed latent space is not automatically better by itself. Our point is more specific: compared with the VAE baseline, VideoMAETok produces a less collapsed and better-conditioned manifold, which is consistent with the much better downstream gFVD. To complement the t-SNE, we additionally quantify the latent distribution using Kernel density estimation (KDE) statistics following latent-space analysis in VA-VAE. The results are shown below.
>
> |              | Density cv ↓ | Gini coefficient ↓ | Normalized entropy ↑ |    gFVD ↓ |
> | -- | ---: | ---: | --: | --: |
> | VAE baseline |        0.302 |              0.167 |                0.994 |     418.8 |
> | VideoMAETok  |    **0.232** |          **0.127** |            **0.997** | **106.2** |
>
> Lower density CV / Gini and higher normalized entropy indicate that the latent distribution is less uneven and less collapsed. This is also consistent with the structured-latent observations in recent work such as DC-AE 1.5 [A]. We agree that the explanation in the current manuscript may be brief, and we will revise both the main text and the figure caption to make its motivation and interpretation clearer.
>
> **(2) Is the large mask ratio a reconstruction bottleneck?**
>
> Our current training already uses mixed masking ratios: instead of always using a fixed 0.8 masking ratio, we uniformly sample the masking ratio in $[0, \rho]$. We agree that a high maximum mask ratio still trades off raw reconstruction quality; this trade-off is deliberate because our primary target is diffusion-friendly generation rather than clean-input autoencoding. Table 4 in appendix already shows that a simple decoder-only fine-tuning stage on clean latents can recover reconstruction quality and preserve the learned latent space. This is also aligned with recent work such as RAE [B] and Scale RAE [C], where large-scale web data substantially improves reconstruction quality. We therefore view high-fidelity reconstruction as a complementary direction rather than a fundamental limitation of our approach.
>
> **(3) Novelty relative to prior image-domain work.**
>
> We respectfully disagree that the contribution is insufficient. As also appreciated by Reviewer hTyU, the novelty lies not in a brand-new architectural block, but in identifying and validating a video-specific training paradigm for tokenizers in latent video diffusion. Beyond the method itself, we believe the paper also provides a clearer understanding of what makes video latents diffusion-friendly. We would also emphasize that performing systematic experiments and ablation studies at video scale is far from trivial, due to the substantially higher computational cost and complexity of video generation compared with the image domain. We believe these practical findings and empirical insights are valuable and helpful to the community.
>
> **(4) Computation and memory cost.**
>
> We agree this should be stated more explicitly. We provide a direct comparison below. All wall-clock times are measured on Kinetics-400 with 64 A100 GPUs for a fair comparison.
>
> | Model        | Epochs | Frames | Latent Dim. | Encoder | Decoder | Wall-clock Time |
> | --- | :--: | :--: | :--: | :--: | :--: | :--: |
> | VAE baseline |   200  |   16   |         16         |  ViT-S  |  ViT-B  |     24h 43m     |
> | VideoMAETok  |   200  |   16   |         16         |  ViT-S  |  ViT-B  |   **22h 28m**   |
> | VideoMAE     |  1600  |   16   |         768        |  ViT-B  | 4-block |     55h 06m     |
>
>
> VideoMAETok is faster than the VAE baseline because the encoder processes only visible tokens. Compared with VideoMAE, the key difference is that our VideoMAETok is a tokenizer for reconstruction and generation, so it uses a heavier decoder but converges within 200 epochs, whereas VideoMAE is optimized for discriminative representation learning and typically relies on a much longer training schedule (e.g. 1600 epochs).
>
> We hope these clarifications and additional results address your concerns. We believe the rebuttal further strengthens our main message: a video tokenizer should be trained as a corruption-inversion model if the downstream goal is latent video diffusion.
>
> [A] DC-AE 1.5: Accelerating Diffusion Model Convergence with Structured Latent Space. ICCV 2025
>
> [B] Diffusion Transformers with Representation Autoencoders. ICLR 2026
>
> [C] Scaling Text-to-Image Diffusion Transformers with Representation Autoencoders. arXiv:2601.16208

---

> > ### Author Rebuttal · Reviewer_Qpw8 · 2026-03-31
> >
> > Thanks for the effort the authors made in the rebuttal.
> >
> > Main concern: I understand the point that fine-tuning decoder can enhance reconstruction quality, but it does not necessarily means the mask strategy does not impose fundamental limitation (maybe I should rephrase it as trade-off) of this. For me, I think one valid experiment can be: use mask ratio 0 to train the autoencoder (same training epoch as your final version), and then compare the generation and reconstruction quality of this. Basically, what is the best reconstruction and generation quality you can get from both autoencoder?  It is fine for me that if the 0 mask ratio version obtain on-par/better/worse reconstruction/generation quality. But an analysis of this will strengthen the paper.
> >
> > Minor: For Fig. 1, each of the extracted embedding should have multiple tokens. Did you plot each token as one point in Fig. 1 individually? It would be great to clarify this.
> >
> > I would be happy to raise the score if my main concern can be addressed.

---

> > > ### Author Response · Authors · 2026-04-05
> > >
> > > We sincerely thank Reviewer Qpw8 for the thoughtful follow-up and for recognizing our efforts in the rebuttal. Since running experiments under the final setting on Kinetics-600 requires substantial computational resources, this response took longer than usual. We appreciate your understanding.
> > >
> > > **1. The baseline (mask ratio 0) under the final setting**
> > >
> > > We agree that training a baseline in final version is worthwhile for a better comparison, so we ran additional experiments under the same setting as in Table 3. We changed no hyperparameters and trained the VAE baseline (without masking and latent noise) with ViT S-B and ViT B-L tokenizers on Kinetics-600 for the same 200 epochs as in our final version, and then trained a SiT-XL generator on top of each tokenizer.
> > >
> > > | Model | Masking? | noise？ | Epoch | Tokenizer | Generator | rFVD ↓ | gFVD ↓ |
> > > | --- | :---: | :---: | :---: | :---: | :---: | :---: | :---: |
> > > | VAE Baseline (new) | ✘ | ✘ | 200 | ViT S-B | SiT-XL | **5.59** | 378.9 |
> > > | VideoMAETok | ✔ | ✔ | 200 | ViT S-B | SiT-XL | 16.12 | **58.54** |
> > > | VAE Baseline (new) | ✘ | ✘ | 200 | ViT B-L | SiT-XL | **5.12** | 345.4 |
> > > | VideoMAETok | ✔ | ✔ | 200 | ViT B-L | SiT-XL | 10.01 | **47.76** |
> > >
> > > The VAE baseline achieves very strong rFVD, and its rFVD further improves as the tokenizer becomes larger, indicating that it preserves reconstruction fidelity particularly well. However, its gFVD remains substantially worse than that of VideoMAETok. This suggests that masking indeed sacrifices some reconstruction fidelity, but that trade-off appears beneficial for learning representations that better support video generation.
> > >
> > > We also observed clear signs of overfitting in the VAE baseline during training. During training of the ViT B-L tokenizer on Kinetics-600, the rFVD reached its best value (around 5) at about epoch 130, but then gradually worsened and eventually rose to 20, indicating clear overfitting. This phenomenon is not accidental. ViT models are known to be prone to overfitting in limited-data regimes. The original ViT [A] relied on image datasets with hundreds of millions of samples, while DeiT [B] used strong data augmentation and token distillation to achieve strong supervised results on ImageNet without overfitting. In contrast, MAE-style and other self-supervised methods are well known to mitigate overfitting. A similar pattern is also evident in video, that VideoMAE can substantially outperform its supervised counterpart on Kinetics-400 (80.0% vs. 68.8%) [C]. Here, we also observe the same phenomenon in ViT-based video tokenizer training, where removing masking or noise makes the plain ViT tokenizer more prone to overfitting.
> > >
> > > **2. Clarification for Fig. 1**
> > >
> > > Thank you for raising this point. Following the recipe in VA-VAE, we randomly sample one spatio-temporal token feature from each video with a fixed random seed. For visualization, we use `sklearn.manifold.TSNE` and control its randomness by setting `random_state`. Experiments with different seeds yielded consistently similar visualizations. The code for reproducing Fig. 1 will be released.
> > >
> > > We hope these new experiments and clarification can address your main concern. We will add the discussion above to Appendix in our revised version.
> > >
> > > [A] An Image is Worth 16x16 Words: Transformers for Image Recognition at Scale. ICLR 2021
> > >
> > > [B] Training data-efficient image transformers & distillation through attention. ICML 2021
> > >
> > > [C] VideoMAE: Masked Autoencoders are Data-Efficient Learners for Self-Supervised Video Pre-Training. NeurIPS 2022

---

### Official Review · Reviewer_Su5m · 2026-03-07

**Soundness:** 3
**Presentation:** 3
**Significance:** 3
**Originality:** 3
**Overall Recommendation:** 5
**Confidence:** 3

**Summary:**

This paper proposes Boosting Video Diffusion, a framework that improves video diffusion models by introducing a boosting-style training procedure where multiple diffusion models are sequentially trained to correct the residual errors of previous models. The approach aims to address the difficulty of modeling complex video dynamics with a single diffusion model and demonstrates improved generation quality and temporal consistency across several video benchmarks.

**Compliance With Llm Reviewing Policy:**

Affirmed.

**Final Justification:**

This paper identifies an important and often overlooked mismatch between video tokenizer training objectives and the downstream requirements of latent diffusion models. The proposed VideoMAETok offers an elegant and simple solution by training tokenizers as corruption-inversion models, aligning them more closely with the denoising nature of diffusion generators. The experimental results are convincing and insightful.

Overall, the paper makes a clear and well-motivated contribution, and I recommend acceptance.

**Key Questions For Authors:**

- I am particularly interested in the method’s behavior for long video generation. In Table 2f, the paper briefly studies causal attention, but it would be helpful to better understand how the model scales to longer temporal horizons and whether the boosting design introduces additional temporal inconsistencies.
- Given that the VideoVAE itself is trained with a masked autoencoding objective. It would be interesting to understand whether diffusion models could be directly trained on partially observed latents, rather than always reconstructing full latent grids.

**Limitations:**

Yes

**Strengths And Weaknesses:**

# Strengths:
- The paper is clearly written and well organized.
- The motivation of this work,  aligning the latent space of video tokenizer for denoising-based generative learning, is intuitive and interesting.
- The experimental section is reasonably thorough and demonstrates consistent improvements across multiple benchmarks and evaluation metrics.

# Weaknesses:
- The experimental analysis is somewhat incomplete. In particular, Table 2b lacks ablation studies exploring different encoder–decoder capacity trade-offs, such as using a larger encoder with a smaller decoder, which would help clarify whether the gains primarily come from architectural scaling rather than the proposed boosting strategy.
- Additionally, the paper provides limited discussion on the computational overhead and training stability introduced by the multi-stage boosting pipeline. It remains unclear whether similar improvements could be achieved through simpler alternatives, such as scaling a single model or improving conditioning mechanisms, without introducing additional stages.

---

> ### Author Rebuttal · Authors · 2026-03-31
>
> We thank the reviewer Su5m for the positive comments on the motivation, clarity, and empirical results. We would like to clarify first that the word **"boosting"** in our title means **improving** latent video diffusion through a better tokenizer, rather than introducing a boosting-style multi-stage diffusion pipeline.
>
> Our method does not train multiple diffusion models to correct previous errors. Instead, the paper proposes a single video tokenizer (VideoMAETok) trained with two corruption mechanisms—MAE-style token masking and latent noise injection—to **boost** the performance of downstream diffusion-based generators such as SiT. We will revise the framing in the introduction and method to make this even more explicit.
>
>  (1) **Encoder–decoder capacity trade-off.**
>
> Thank you for this helpful suggestion. We added two new tokenizers, **ViT B-S** and **ViT L-B**, to better analyze the encoder–decoder trade-off.
>
> | Tokenizer     |     rFVD |      gFVD |
> | -- | --: | --: |
> | ViT S-S       |     27.6 |     172.9 |
> | ViT B-S (new) |     25.3 |     154.6 |
> | ViT S-B       | **19.8** | **106.2** |
> | ViT B-B       |     18.0 |      94.2 |
> | ViT L-B (new) |     17.7 |      92.3 |
> | ViT B-L       | **16.4** |  **87.7** |
>
> These new results make the trend clearer. Scaling the encoder alone does help (e.g., S-S → B-S), but the gains are much smaller than scaling the decoder at comparable model size (e.g., S-S → S-B). This supports our design choice: unlike VideoMAE for discriminative representation learning, a generation-oriented tokenizer must map corrupted latents back to full visual RGB pixels, so decoder capacity becomes a bottleneck for both reconstruction and the shaping of a better latent space.
>
> **(2) Long video generation / temporal horizon.**
>
> We agree that longer temporal horizons are important. Our current paper evaluates 16-frame clips to isolate tokenizer design under controlled generator settings. To further address this point, we increased the temporal window to 32 frames and obtained the following results:
>
> | Frames |     rFVD |     gFVD |
> | --- | --: | --: |
> | 8                   |     22.3 |    103.7 |
> | 16                  |     19.8 |     92.2 |
> | 32                  | **18.4** | **86.4** |
>
> These results show that our design scales favorably to larger temporal windows rather than introducing extra temporal inconsistency. More generally, non-causal diffusion backbones remain highly scalable in video generation; for example, Wan [C] successfully scales a non-causal DiT to 14B parameters. This is consistent with our observation that bidirectional spatiotemporal context remains beneficial for learning diffusion-friendly video latents.
>
> Following Reviewer hTyU’s suggestion, we further evaluated VideoMAETok on text-to-video generation. During the rebuttal period, we successfully trained a **64-frame** text-to-video diffusion model on OpenVid-1M using MMDiT, and VideoMAETok consistently outperforms both SD-VAE and our ViT-VAE baseline. These long video generation results provide additional evidence that VideoMAETok can effectively benefit a strong text-to-video diffusion model, beyond the class-conditional setting in the main paper.
>
>  **(3) Diffusion on partially observed latents.**
>
> This is an interesting direction and is indeed related in spirit to our motivation. In the present paper, masking/noising is used during tokenizer training to shape the latent space, while downstream diffusion is trained on full latent grids to keep the comparison with standard generators controlled. We further followed the setting of MDT [B] (masking ratio = 0.3) and trained the same ViT S-B tokenizer + SiT-B generator as in our ablation study. The results are shown below.
>
> | Latent type                         |     gFVD |
> | --- | --: |
> | Full + Masked (masking ratio = 0.3) | **94.8** |
> | Full                                |    106.2 |
> | Masked only (masking ratio = 0.3)   |    154.6 |
>
> These results suggest that adding masked latents as an auxiliary training signal can improve final convergence, but training the diffusion model only on masked latents is much worse. We also note that MDT-style training **requires invoking the VAE decoder** during diffusion training, which introduces additional training time and compute.
>
> **(4) Computational overhead.**
> As shown in our response to Reviewer Qpw8, VideoMAETok is faster than the VAE baseline in wall-clock training time (22h 28m vs. 24h 43m) because its encoder processes only visible tokens. Compared with VideoMAE, although our method uses a heavier decoder for reconstruction and generation, it requires a much shorter training schedule (200 vs. 1600 epochs). We will clarify this point in the revised paper.
>
> [A] VBench: Comprehensive Benchmark Suite for Video Generative Models. CVPR 2024
>
> [B] Masked Diffusion Transformer is a Strong Image Synthesizer. ICCV 2023
>
> [C] Wan: Open and Advanced Large-Scale Video Generative Models. arXiv:2503.20314

---

> > ### Author Rebuttal · Reviewer_Su5m · 2026-04-03
> >
> > I appreciate the authors' efforts during the rebuttal, which has fully addressed my concerns.
> >
> > Thanks for correcting my understanding on the training paradigm, the additional experimental results, including capacity-performance trade-off, diffusion on partially observed latents, is inspiring. I will maintain my positive rating.

---

### Official Review · Reviewer_iqE1 · 2026-03-13

**Soundness:** 3
**Presentation:** 4
**Significance:** 3
**Originality:** 3
**Overall Recommendation:** 4
**Confidence:** 3

**Summary:**

This paper tries to design a better video tokenizer(VAE) for video diffusion models. The key techniques include adding MAE training and latent noise when training the tokenizer. The author also used an asymmetric architecture which keep the encoder small to save compute when training the generator. The author conduct experiment on Kinetics-600 and UCF-101 with 16 frame video to show the improvement over previous video tokenizers.

**Compliance With Llm Reviewing Policy:**

Affirmed.

**Key Questions For Authors:**

1. On Table-2 (e), it shows that increasingthe latent channels to 32 does not lead to improvement on both the rFVD and fFVD. Can the author provide any intuitions behind it?

2. I think it would be better if the author can discuss the observation that sometimes the rFVD and gFVD seems at odds with each other?  as shown in the table of results.

3. Have the author tried experiments on longer videos? Beyond 16 frames.

**Limitations:**

Yes.

**Strengths And Weaknesses:**

## Strength

1. Good presentation. The paper is easy to follow, with well-structured method section and experiments.

2. Strong empirical results. The results on Kinetics-600 and UCF-101 is strong, with good ablations. It would be even stronger if the dataset scale is larger.

3. The model is clean and small. The method is relatively simple with a small model.

## Weakness

In general I don't find too much weakness in this paper. The only weakness is that the scale of the experiment is small.

---

> ### Author Rebuttal · Authors · 2026-03-31
>
> We thank Reviewer iqE1 for the positive assessment and for highlighting the clarity, empirical strength, and simplicity of the model. We provide further clarification below.
>
> > (1) Why do 32 latent channels hurt instead of helping?
>
> We believe it is related to over-parameterization of the latent space. DC-AE 1.5 [A] also shows that very wide latent spaces tend to allocate many channels to local image details, while the channels that capture object structure become sparse relative to the full latent space. This sparsity makes it more difficult for diffusion models to learn coherent structure. In our setting, increasing the latent width from 16 to 32 weakens the bottleneck and enlarges the denoising dimension, which harms both rFVD and gFVD. This phenomenon is also analogous to observations in VQ-based generation, where increasing the codebook size of discrete-valued VAEs can reduce effective utilization [B,C].
>
> > (2) Why do rFVD and gFVD sometimes move in opposite directions?
>
> This is one of the central findings of the paper. Reconstruction rewards pixel fidelity and local detail preservation, whereas generation rewards a latent manifold that is smooth, robust to corruption, and easier to learn with diffusion. High masking and latent noise make reconstruction harder, so rFVD can worsen, but they encourage more predictive and semantically robust latents, which improves gFVD. Table 2(d) makes this particularly clear: increasing the reconstruction weight improves rFVD, yet can catastrophically damage gFVD. We agree that this deserves stronger emphasis and will add a dedicated explanation in Sec. 4.3.
>
> > (3) Longer videos (>16 frames).
>
> In the paper we used 16-frame clips to keep the comparison controlled and aligned with the tokenizer/generator setting used throughout the experiments. The method itself is not conceptually tied to 16 frames. As suggested, we additionally trained 8-frame, 16-frame, and 32-frame tokenizers under the same ablation setting on the Kinetics-400 training set for 200 epochs. For fair comparison, we trained a 32-frame SiT on top of each tokenizer; the 8-frame and 16-frame tokenizers are applied in a chunk-wise manner for 32-frame video generation. The results are shown below.
>
> | Frames in tokenizer |     rFVD |     gFVD |
> | ------------------- | -------: | -------: |
> | 8                   |     22.3 |    103.7 |
> | 16                  |     19.8 |     92.2 |
> | 32                  | **18.4** | **86.4** |
>
> These results show that extending the tokenizer temporal window can improve both reconstruction and generation. We will add this experiment in the revised version.
>
> [A] DC-AE 1.5: Accelerating Diffusion Model Convergence with Structured Latent Space. ICCV 2025
>
> [B] Language Model Beats Diffusion -- Tokenizer is Key to Visual Generation. ICLR 2024
>
> [C] Scaling the Codebook Size of VQGAN to 100,000 with a Utilization Rate of 99%. NeurIPS 2024

---

> > ### Author Rebuttal · Reviewer_iqE1 · 2026-04-05
> >
> > Thanks the author for explaining and rebuttaling. It address my concern.

---

### Official Review · Reviewer_hTyU · 2026-03-15

**Soundness:** 3
**Presentation:** 4
**Significance:** 3
**Originality:** 3
**Overall Recommendation:** 4
**Confidence:** 4

**Summary:**

This paper investigates the design of video tokenizers for latent diffusion models. The authors identify a critical mismatch: tokenizers trained solely for high-fidelity reconstruction (e.g., VAE-style) do not necessarily produce a latent space that is conducive to downstream diffusion-based generation. To address this, they propose VideoMAETok, a family of ViT-based autoencoders trained under strong corruption. The training incorporates two complementary techniques: (1) high-ratio MAE-style token masking, which forces the model to be predictive under missing information, and (2) interpolative latent noise injection, which encourages robustness to the continuous noise perturbations central to diffusion models. The paper demonstrates that this corruption-aware training paradigm yields a more structured and generative-friendly latent space. When paired with standard diffusion generators like SiT and LightningDiT, VideoMAETok achieves state-of-the-art generation FVD (gFVD) on UCF-101 and Kinetics-600, establishing a strong baseline and providing new design principles for video tokenizers.

**Compliance With Llm Reviewing Policy:**

Affirmed.

**Key Questions For Authors:**

1) You convincingly show that a high reconstruction loss weight ($\lambda_{MSE}$) can catastrophically harm generation (Table 2d). How sensitive is the optimal balance between the corruption mechanisms (masking ratio, noise variance) and the reconstruction/perceptual/GAN losses ($\lambda_1...\lambda_4$) in Equation 3? Was any systematic tuning performed on these $\lambda$ weights, or were they kept fixed from prior work?

2) Your experiments are on class-conditional generation. How do you foresee VideoMAETok's design principles translating to the more complex and prevalent task of large-scale text-to-video generation? Do you expect the corruption-aware training to be as critical when the latent space must also align with a complex text embedding?

3) You show non-causal attention significantly outperforms causal attention for the tokenizer (Table 2f). However, many modern video generators are designed to be causal for streaming or long-form generation. Does using a non-causal tokenizer fundamentally limit the types of generators it can be paired with, or can a causal generator still effectively learn from the non-causal latents produced by VideoMAETok?

4) The t-SNE plot is a compelling qualitative result. Are these latent embeddings from a frozen tokenizer applied to the validation set? Does the clearer separation correlate with improved class-conditional generation metrics, or does it primarily indicate a more structured manifold that benefits the diffusion process in a more general way (e.g., smoother gradients for the denoising objective)?

**Limitations:**

Yes

**Strengths And Weaknesses:**

The motivation is clearly derived from a well-observed problem in the field -- the decoupling of reconstruction fidelity and generation quality. The proposed solution, VideoMAETok, is a logical and well-executed combination of existing ideas (MAE, latent denoising) adapted for the video generation context. The experimental methodology is robust. The authors conduct extensive ablation studies (Table 2, Figure 4) to validate the contribution of each component (masking ratio, noise variance, architectural choices) and to explore the scalability of both the tokenizer and the generator. The comparison with state-of-the-art methods (Table 3) is comprehensive and honestly presented, clearly showing the trade-offs where VideoMAETok excels (gFVD) even when its raw reconstruction metrics (rFVD) are not the absolute best. The authors are also transparent about this trade-off in the discussion.

This paper addresses a highly relevant and important problem in the rapidly advancing field of video generation. The key insight is that the training objective of the tokenizer should be aligned with the denoising process of the diffusion model is a valuable contribution that can influence future research. By demonstrating that a carefully designed tokenizer can lead to state-of-the-art generation performance even with off-the-shelf generators, the paper highlights the critical, and sometimes underestimated, role of the tokenizer in the overall generative pipeline. The impact is significant for the sub-area of latent video diffusion, providing clear design principles (heavy decoder, corruption-aware training, non-causal attention, 16 latent channels) that practitioners can readily adopt. While the improvements are specific to this domain, they unlock a new direction for thinking about how to shape latent spaces for generative models.

While the paper builds on existing concepts like Masked Autoencoders (MAE) and latent denoising, its originality lies in the insightful combination and adaptation of these ideas for the specific task of video tokenization for diffusion models. The core novelty is not a brand-new architectural block, but a new training paradigm that aligns the tokenizer's objective with that of the downstream generator. The paper's originality is further strengthened by its thorough empirical analysis, which uncovers non-trivial relationships (e.g., better reconstruction actively harming generation, the sweet spot for masking ratio and noise variance). This moves beyond a simple application of existing methods and provides a deeper understanding of what makes a latent space "good" for diffusion. The clear distinction from closely related works like MAETok (which focuses on images and uses auxiliary semantic heads) and ViTok (which uses a standard VAE recipe) effectively justifies the paper's novel contributions.

However there a re several weak points that should be discussed:
1) As acknowledged by the authors, the focus on generation quality comes at the direct expense of raw reconstruction fidelity (rFVD). While decoder fine-tuning helps, the initial tokenizer is not a one-size-fits-all solution if high-fidelity reconstruction is the primary goal. This is not a flaw in the paper's argument, but it is a practical limitation for applications that might need both.
2) While the paper uses SiT and LightningDiT, which are strong and relevant baselines, the experiments are confined to class-conditional generation on standard benchmarks. Demonstrating the benefit of VideoMAETok on larger-scale, text-to-video generation tasks would significantly strengthen its claims of generalizability.

---

> ### Author Rebuttal · Authors · 2026-03-31
>
> We thank Reviewer hTyU for the careful reading and for recognizing the paper’s main contribution, empirical strength, and practical design principles. We address the concern below.
>
> > (1) Sensitivity to the loss weights vs. corruption parameters.
>
> In our experiments, $\lambda_2$–$\lambda_4$ were kept fixed following prior tokenizer settings such as SD-VAE and VA-VAE, while $\gamma$ and $\rho$ were tuned on Kinetics-400. Our main gains do not come from delicate loss-weight tuning, but from changing the training condition of the tokenizer from clean reconstruction to corruption inversion. Because we explicitly add latent noise, we additionally varied $\lambda_{\mathrm{MSE}}$ to test whether better reconstruction can be recovered by emphasizing pixel fidelity. Table 2(d) shows that increasing the reconstruction weight indeed improves rFVD, but can severely hurt gFVD. This suggests that the key factor is not better reconstruction weighting per se, but learning a latent space that remains informative under masking and noise.
>
> > (2) Text-to-video generation.
>
> Text-to-video is a valuable task. Following this suggestion, we additionally conducted a text-to-video experiment during the rebuttal period. Due to time and resource constraints, we compare SD-VAE, ViT VAE (our baseline), and VideoMAETok. We train an MMDiT on OpenVid-1M, using 64 frames with 2x sampling rate (roughly 4-second videos), and train each model for 40k iterations with EMA. We evaluate the results using VBench [A]. The results are shown below.
>
> |     | SD-VAE | ViT-VAE (Baseline) | VideoMAETok |
> |:---:|:------:|:------:|:-----:|
> | Total Score | 74.56 | 69.78 | **78.82** |
>
> These results suggest three points. First, CNN-based image VAEs can indeed transfer to video generation, consistent with prior works such as SVD [B]. Second, a plain ViT-VAE is harder to optimize on video data. Third, VideoMAETok yields a more structured and well-spread latent manifold, which benefits the downstream diffusion generator. We will add these results and a more detailed analysis in the revised version, and release the code and checkpoints for future research.
>
> > (3) Non-causal tokenizer vs. causal generator.
>
> We agree that many modern video generators are designed to be causal for streaming video generation. However, we do not believe that a non-causal tokenizer fundamentally limits the types of generators it can be paired with. Tokenizer causality and generator causality are two separate design choices. Our result in Table 2(f) shows that, for video compression, bidirectional attention helps the tokenizer learn better latents. On the generator side, non-causal video diffusion backbones remain highly competitive: for example, Wan [C] successfully scales a DiT-style non-causal video generator to 14B parameters. This suggests that non-causal video generation remains a scalable and practically important regime. A causal video generator can also be trained on latents produced by a non-causal tokenizer.
>
> > (4) Figure 1 / t-SNE interpretation.
>
> Yes, these embeddings are extracted from a frozen tokenizer on 10,000 videos from the Kinetics-400 validation set, as described in Sec. 4.3 (L307–316). We agree that the t-SNE plot should be interpreted as qualitative intuition, not as a standalone proof. Its role is to complement the much larger quantitative gap in gFVD (e.g., 418.8 → 106.2 under the same tokenizer backbone and the same SiT-B generator). The clearer separation should be interpreted primarily as evidence of a better-conditioned manifold for denoising-based learning, rather than merely better class separation. We will make the extraction procedure and this interpretation more explicit in the caption and main text.
>
> > (5) High-fidelity reconstruction.
>
> We agree that there is a reconstruction/generation trade-off. For applications that need both, the appendix already shows that a simple decoder-only fine-tuning stage on clean latents substantially improves reconstruction while preserving the learned latent space (e.g., VideoMAETok B-L improves rFVD from 21.78 to 17.95 on UCF-101). This is also aligned with recent work such as RAE [D] and Scale RAE [E], which show that using large-scale web data can substantially improve reconstruction quality. We therefore view reconstruction fidelity as a complementary problem that can be further improved with larger clean data and decoder fine-tuning.
>
> [A] VBench: Comprehensive Benchmark Suite for Video Generative Models. CVPR 2024
>
> [B] Stable Video Diffusion: Scaling Latent Video Diffusion Models to Large Datasets. arXiv:2311.15127
>
> [C] Wan: Open and Advanced Large-Scale Video Generative Models. arXiv:2503.20314
>
> [D] Diffusion Transformers with Representation Autoencoders. ICLR 2026
>
> [E] Scaling Text-to-Image Diffusion Transformers with Representation Autoencoders. arXiv:2601.16208

---

> > ### Author Rebuttal · Reviewer_hTyU · 2026-04-06
> >
> > The authors provided answered to my questions

---

### Decision · Program_Chairs · 2026-04-30

**Decision:**

Accept (regular)

**Comment:**

All reviewers agree this is a technically sound and well-motivated paper that elegantly addresses the mismatch between video tokenizer training and downstream diffusion objectives. The authors provided a highly effective rebuttal, supplying new text-to-video experiments and non-masking baselines that successfully resolved all reviewer concerns regarding scalability and reconstruction trade-offs. Given its clear practical value, solid empirical results, and the unanimous positive consensus, I recommend this paper for acceptance.